# Machine learning-assisted discovery of growth decision elements by relating bacterial population dynamics to environmental diversity

**Honoka Aida, Takamasa Hashizume, Kazuha Ashino, Bei-Wen Ying\***

School of Life and Environmental Sciences, University of Tsukuba, Tsukuba, Japan

**Abstract** Microorganisms growing in their habitat constitute a complex system. How the individual constituents of the environment contribute to microbial growth remains largely unknown. The present study focused on the contribution of environmental constituents to population dynamics via a high-throughput assay and data-driven analysis of a wild-type *Escherichia coli* strain. A large dataset constituting a total of 12,828 bacterial growth curves with 966 medium combinations, which were composed of 44 pure chemical compounds, was acquired. Machine learning analysis of the big data relating the growth parameters to the medium combinations revealed that the decision-making components for bacterial growth were distinct among various growth phases, e.g., glucose, sulfate, and serine for maximum growth, growth rate, and growth delay, respectively. Further analyses and simulations indicated that branched-chain amino acids functioned as global coordinators for population dynamics, as well as a survival strategy of risk diversification to prevent the bacterial population from undergoing extinction.

## Editor's evaluation

In this manuscript, the authors quantitatively analyze the growth curves for *E. coli* under a large number of growth conditions and use different machine learning methods to tackle the combinatorial complexity of conditions as well as to predict growth parameters from media composition. The large datasets and the use of ML to handle such complex modeling will be of general interest to the biology community.

**\*For correspondence:**
ying.beiwen.gf@u.tsukuba.ac.jp

## Introduction

Highly diversified microorganisms grow in highly differentiated habits (*Escalas et al., 2019*; *Levin et al., 2021*). The measurement of diversity in both genetics and the environment is essential to understand community outcomes as an ecological cause and/or consequence (*Shade, 2017*) and the evolutionary and responsive strategies constrained by the environment (*Celani and Vergassola, 2010*; *Fraebel et al., 2017*). To date, studies have focused more on genetic diversity, e.g., metagenomics (*Handelsman, 2004*; *Chistoserdova, 2010*) and microbial communities (*Mitri and Foster, 2013*; *Heinken et al., 2021*), than on environmental diversity, despite the high complexity of both microbes and environments (*Langenheder et al., 2010*; *Pacheco et al., 2021*). It remains unclear how the individual constituents of the environment (i.e. habitat) contribute to the population dynamics of the microbe or community. Mimicking the environmental diversity in the laboratory by reconstituting the environment (e.g. medium) with known components of defined amounts or magnitudes might be applicable to address this issue.

Microbial population dynamics are commonly represented by growth curves (*Egli, 2015*; *Zwietering et al., 1990*; *Tonner et al., 2017*). How a microbial population (species) fits the habitat (environment) has largely been evaluated by three parameters derived from the growth curve, i.e., the lag time, growth rate, and saturated population size, which quantitatively represent the lag, exponential, and stationary phases of the growth curve, respectively (*Blomberg, 2011*; *Peleg and Corradini, 2011*). The lag time has been reported to be crucial for bacterial growth under environmental stress (*Zhou et al., 2011*; *Guillier et al., 2005*). The growth rate has been associated with proteome allocation (*Scott et al., 2010*; *Zhu and Dai, 2019*), ribosome function (*Gourse et al., 1996*; *Dai and Zhu, 2020*), and gene expression (*Nilsson et al., 1984*; *Klumpp et al., 2009*); thus, it represents the adaptiveness (fitness) of the microbial population (*Towbin et al., 2017*; *Saether and Engen, 2015*). The three growth parameters are likely coordinated with each other. Previous studies have observed trade-offs between the growth rate and either the population size (*Novak et al., 2006*; *Engen and Saether, 2006*) or the lag time (*Basan et al., 2020*) within identical species, as well as correlated changes in the growth rate and saturated population size among genetically diversified strains (*Liu et al., 2006*; *Nishimura et al., 2017*). Whether and how environmental diversity affects the three growth parameters remain unknown.

To address these questions, a quantitatively high-throughput survey linking growth parameters to environmental diversity is required. As the microbial population dynamics have been shown to be strongly dependent on the growth medium (*Egli, 2015*), relating the bacterial growth profile to the medium constitution is applicable to address the issue. Recently, both high-throughput technologies for bacterial growth analysis (*Blomberg, 2011*) and data-driven computational approaches have been developed for studying complex systems (*Jordan and Mitchell, 2015*; *Gilpin et al., 2020*; *Xu and Jackson, 2019*). In particular, machine learning (ML) techniques have been widely applied to studies on genetic diversity (*Schrider and Kern, 2018*; *Libbrecht and Noble, 2015*), metabolic engineering

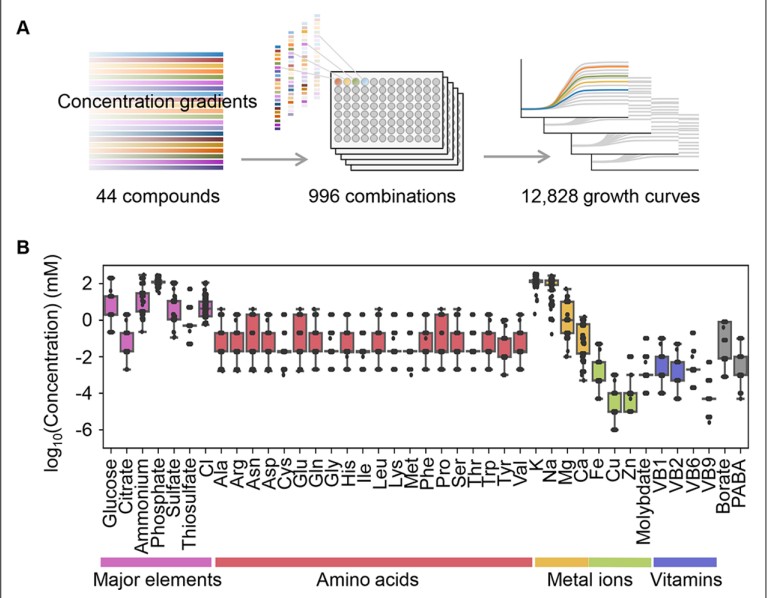

**Figure 1.** Relating bacterial growth to environmental diversity. (**A**) Flowchart of experimental conditions and data attainment. Colour gradation indicates the concentration gradient of the pure chemical compound used in the medium combinations. (**B**) Concentration variation of the components comprising the medium combinations. Colour variation indicates the categories of elements. The concentrations are indicated on a logarithmic scale.

The online version of this article includes the following figure supplement(s) for figure 1:

**Figure supplement 1.** Variations in concentration gradients of the components.

**Figure supplement 2.** Experimental tests of the changes in growth rate (*r*) responding to the concentration gradients of the minor compounds.

**Figure supplement 3.** Experimental tests of the relationship between saturated population density (*K*) and the minor compounds.

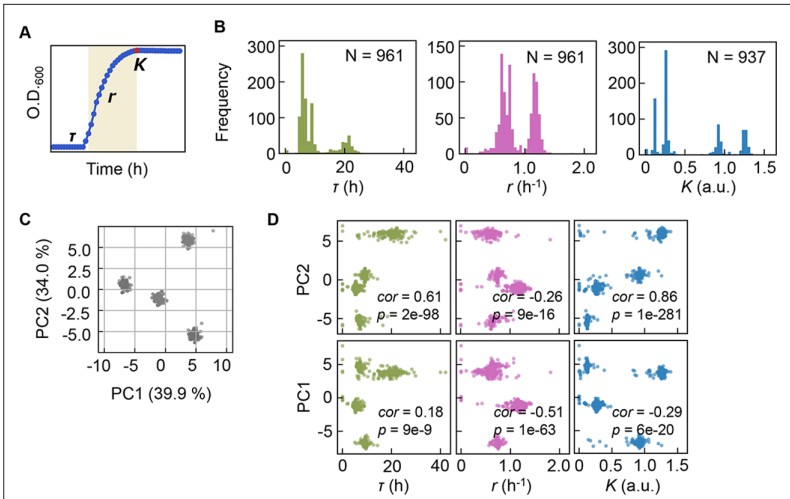

**Figure 2.** Bacterial growth profiling. (**A**) Three growth parameters calculated from growth curves. The lag time (τ), the growth rate (r), and the saturated population size (K) are indicated. (**B**) Distributions of the three parameters. The numbers of medium combinations (**N**) used are indicated. (**C**) Principal component analysis (PCA) of medium combinations. The contributions of PC1 and PC2 are shown. (**D**) Correlations of the three parameters to PC1 and PC2. Spearman's correlation coefficients and the p values are indicated.

The online version of this article includes the following source data and figure supplement(s) for figure 2:

**Source data 1.** Medium combinations used in the growth assay.

**Figure supplement 1.** Clustering of the medium combinations.

**Figure supplement 2.** Illustration of the fitness landscape.

**Figure supplement 3.** Subculture of the *E. coli* cell population used in the growth assay.

---

(*Kim et al., 2020*), and population dynamics (*Campos et al., 2018*; *Hiura et al., 2021*; *Ashino et al., 2019*). Combining ML approaches with the use of high-throughput measurements of a well-known microbe in well-defined environments has become practical for comprehensive quantitative evaluation of the contribution of environmental factors (e.g. chemical compositions of the habitats) to microbial population dynamics (e.g. bacterial growth). In the present study, a large dataset describing the bacterial population dynamics in a broad environmental gradient of largely varied combinations was experimentally acquired under well-controlled laboratory conditions. ML prediction and niche broadness analysis of the big data linking bacterial growth to environmental diversity (i.e. medium combinations) were performed. The bacterial growth strategy was investigated by means of data-driven approaches.

## Results
### Relating bacterial growth to environmental diversity

Precise bacterial growth profiling was performed by a high-throughput growth assay in varied medium combinations (*Figure 1A*), which were prepared with 44 pure chemical substances that are commonly used in different microbial culture media. As the chemical substances are ionized in solution, these medium combinations finally comprised 41 components (e.g. metal ions, amino acids [AAs], etc.) whose concentrations varied broadly on a logarithmic scale (*Figure 1B*). In brief, a total of 12,828 growth curves of *Escherichia coli* BW25113 grown in 966 different medium combinations were acquired. Three parameters, the lag time (τ), maximal growth rate (r), and saturated population density (K), were subsequently calculated according to the growth curves, which represented the quantitative features in the lag, exponential, and stationary growth phases, respectively (*Figure 2A*). The averaging of the biological replications and the removal of the unreliable measurements finally resulted in 961, 961, and 937 values of τ, r, and K, respectively (*Figure 2B*, *Figure 2—source data 1*). The three parameters all presented multimodal distributions in response to environmental variation, which agreed well

**Figure 3.** Evaluation of machine learning (ML) models. (**A**) Workflow of ML. (**B**) Accuracy of the ML models. Boxplots of the evaluation metrics obtained in the ML prediction of growth rate are shown. The root mean squared errors (RMSEs) of five independent tests are indicated as black points.

The online version of this article includes the following figure supplement(s) for figure 3:

**Figure supplement 1.** Accuracy of the machine learning (ML) and multiple regression models.

**Figure supplement 2.** Time required for the machine learning (ML) model training.

**Figure supplement 3.** Effect of the abundance of training data on the accuracy of gradient-boosted decision tree (GBDT).

**Figure supplement 4.** Accuracy of the machine learning ML models varied with the experimental errors of the data for training.

with the rugged fitness landscapes proposed for adaptive evolution (*Neidhart et al., 2014*) and the immune response (*Kauffman and Weinberger, 1989*).

Clustering analyses and principal component analysis (PCA) were applied to the medium combinations. The 966 medium combinations could be mainly divided into four clusters (*Figure 2C*), roughly with respect to the multimodality of the distributions (*Figure 2—figure supplement 1*). If the three parameters of $\tau$, $r$, and $K$, which all showed the multimodal distributions, were independent, more than eight clusters were anticipated. Only four separate clusters were identified, indicating that the growth parameters were somehow dependent. The three parameters were all correlated with the two main PCs (*Figure 2D*), suggesting that bacterial growth was determined by certain common components comprising medium combinations. The results presented an overview of the relationship between the medium combinations and bacterial growth and indicated the growth law in common mediated by the medium components.

## Decision-making components for bacterial growth

ML approaches were applied to predict the three parameters according to the medium combinations (*Figure 3A*). As a preliminary test, five representative ML models and an ensemble model were trained and evaluated. The results showed that the prediction accuracy was approximately equivalent among the six ML models (*Figure 3B*), independent of the evaluation metrics (*Figure 3—figure supplement 1*). It indicated that the simple ML models were available to tackle the large dataset generated by the throughput growth assay and appropriate for the prediction of bacterial growth according to the environmental details, e.g., medium composition. To determine an explainable linkage between bacterial growth and the medium constitution, the ML model of the gradient-boosted decision tree (GBDT) was chosen for further investigation.

Intriguingly, repeated GBDT prediction showed that the growth parameters were largely determined by a single component out of 41 components comprising the medium (*Figure 4A*, *Figure 4—source data 1*). It seemed that a few key components played a determinant role in bacterial growth. The top 10 features (i.e. components) contributing to the three parameters somehow overlapped (e.g. K, Na, and phosphate), which might reflect the common effect of osmotic balance resulting from these components. Nevertheless, the components of the highest priority in governing the three parameters were highly differentiated, i.e., serine, sulfate, and glucose for $\tau$, $r$, and $K$, respectively (*Figure 4A*). This finding was confirmed by the correlations of the three parameters to the changes in the concentrations of the three components, irrespective of the large variation in other components

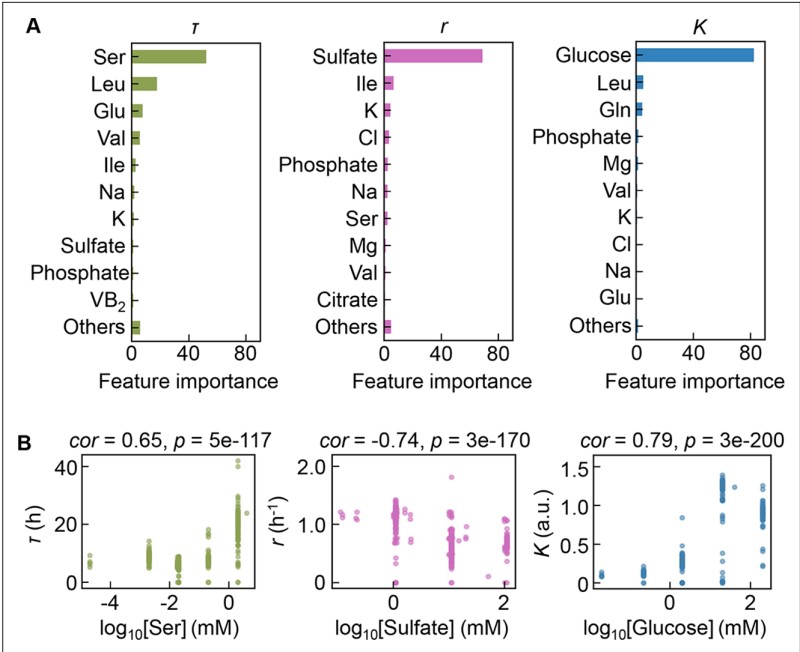

**Figure 4.** Contribution of the components to bacterial growth. (**A**) Relative contributions of the components to the three parameters predicted by gradient-boosted decision tree (GBDT). 10 components with large contributions to the three parameters of lag time (*τ*), growth rate (*r*), and saturated population size (*K*) are shown in order. The remaining 31 components are summed as 'Others'. (**B**) Correlation of the concentrations of the components with the growth parameters. The components with the largest contributions to the three parameters *τ*, *r*, and *K* are shown individually. Spearman's correlation coefficients and the p values are indicated.

The online version of this article includes the following source data and figure supplement(s) for figure 4:

**Source data 1.** Summary of the feature importance of the components for $\tau$, r, and K.

**Figure supplement 1.** Violin plots of the growth parameters at varied ranges of chemical concentrations.

**Figure supplement 2.** Separation of the multimodal distribution of growth rate (*r*).

**Figure supplement 3.** Separation of the multimodal distribution of saturated population size (*K*).

**Figure supplement 4.** Separation of the multimodal distribution of lag time (*τ*).

present (*Figure 4B*, *Figure 4—figure supplement 1*). Since the Spearman's rank correlation was used, the large size of dataset led to high significance, which was somehow discrepant to the graphics. It suggested that growth decisions were highly constrained by a few components and were largely distinguished in response to the growth phase.

## Sensitive components affecting bacterial growth

As the three growth parameters were somehow determinatively decided by a few components, the changes in growth parameters in response to the concentration gradient of each component were evaluated according to the previous study (*Kurokawa et al., 2021*). Here, the area (i.e. the shadowed space, *S*) above the fitting curve of cubic polynomial regression to the normalized plot was newly defined, in which the maxima of both the concentration gradients and the growth parameters were rescaled to one unit (*Figure 5A*, *Figure 5—figure supplement 1*). An assortment of fitting curves was acquired for the target component (*Figure 5—figure supplements 2–4*) because of the various combinations of the remaining 40 components (*Figure 5B*). The mean of these *S* values was calculated and designated the sensitivity of the component for bacterial growth in response to the alternative combinations of other components. A larger value of *S* indicated a higher sensitivity of the component, i.e., indicated larger changes in the growth parameters due to the variation in the concentration gradients of the other 40 components. Consequently, a total of 41 *S* values were acquired with respect to the three parameters, i.e., $S_{\tau}$, $S_r$, and $S_K$ (*Figure 5—figure supplement 5*, *Figure 5—source data 1*), which all presented long-tailed distributions (*Figure 5C*). The sum of $S_{\tau}$, $S_r$, and $S_K$, which was defined

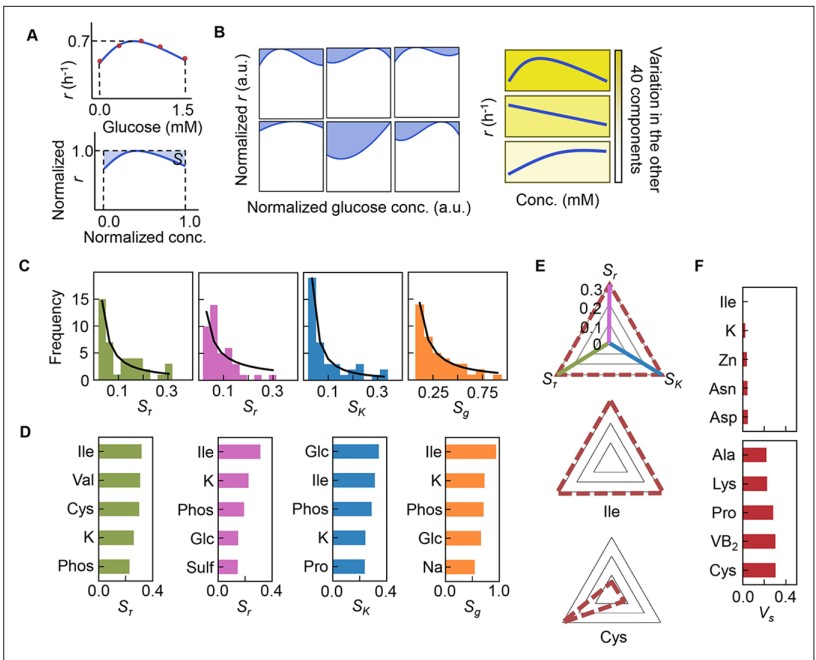

**Figure 5.** Sensitivity of the components. (**A**) Definition of sensitivity. As an example, the upper and bottom panels indicate the regression curve across the concentration gradient of glucose and the normalized regression curve, in which both the concentration gradient and the growth rates are rescaled within one unit, respectively. The shaded area was determined as the sensitivity (**S**) of glucose. (**B**) Variation in the sensitivity. Six different regression curves, i.e., six different $S$ values, of glucose are shown, which result from the alternative combinations of the other 40 components (left panels). The yellow gradation and blue lines represent the variation in medium combinations and the corresponding regression curves, respectively (right panels). (**C**) Distributions of the mean sensitivities. The mean $S$ values evaluated according to lag time ($\tau$), growth rate ($r$), and saturated population size ($K$) are shown as $S_\tau$, $S_r$, and $S_K$, respectively. The sum of the three $S$ values is shown as global sensitivity ($S_g$). The black lines indicate the fitting curves of the power law. (**D**) Most sensitive components. The components with the largest $S$ values are shown in the order of value. (**E**) Balance of sensitivity. The balance of sensitivity is visualized by the triangle of $S_r$, $S_K$, and $S_\tau$ in red dotted lines. The solid lines in pink, blue, and green represent $S_r$, $S_K$, and $S_\tau$, respectively. Those close to or far from an equilateral triangle are determined as the balanced (Ile) or biased (Cys) sensitivity in response to the growth phases, respectively. (**F**) Variance of sensitivity. The components with either the smallest or the largest $V_s$ are shown in the order of value. Five components of either balanced or biased sensitivity are shown.

The online version of this article includes the following source data and figure supplement(s) for figure 5:

**Source data 1.** Summary of sensitivity.

**Figure supplement 1.** Schematic drawing of the analytical procedure of $S$.

**Figure supplement 2.** Normalized regression curves of the growth rates ($r$).

**Figure supplement 3.** Normalized regression curves of the saturated population size ($K$).

**Figure supplement 4.** Normalized regression curves of the lag time ($\tau$).

**Figure supplement 5.** Sensitivity of the components.

as the global sensitivity ($S_g$) of the component across the three growth phases, showed a similar long-tailed distribution shape. The four distributions were all likely to follow the power law (**Evans et al., 2021**; **Furusawa and Kaneko, 2006**), which agreed well with the ML-predicted conclusion that only a few components determined the growth. This finding strongly suggested that the decision-making components for bacterial growth were present among the 41 components, regardless of the complex interactions among these components.

The components with the largest $S$ values, i.e., Ile, K, and phosphate (**Figure 5D**), overlapped among the three parameters, suggesting that these components were highly sensitive to the fluctuation of other components for all growth phases. In particular, Ile was the most sensitive component, as it presented the largest $S_g$, i.e., the largest changes in bacterial growth responding to the concentration gradient of Ile in different combinations of other chemicals (i.e. patterns shown in **Figure 5—figure**

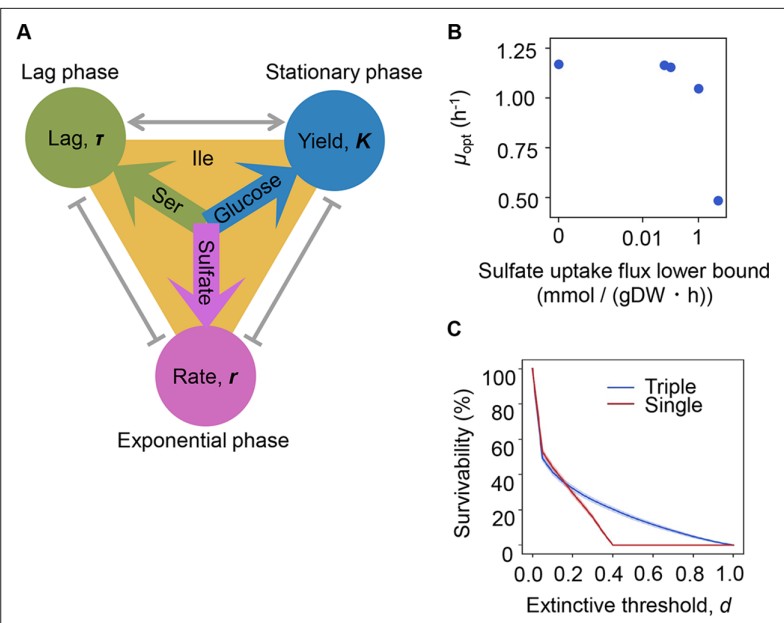

**Figure 6.** Growth strategy of risk diversification. (**A**) Schematic drawing of the decision-making components for bacterial growth. (**B**) Flux balance analysis (FBA) simulation. The predicted growth rates are plotted against the input rate of sulfate uptake. (**C**) Theoretical simulation of survival probability. The blue and red lines represent the growth strategies of the multiple and single decision makers, respectively. The shading covering the red and blue lines indicates the SD.

*supplement 1*). The result well supported the finding that the components dominating the three second-priority parameters were Leu and Ile (*Figure 4A*). Additionally, analysing the variance ($V_s$) of the $S_r$, $S_r$, and $S_K$ values showed that the largest $V_s$ values was in Cys (*Figure 5F*, *Figure 5—figure supplement 5*), indicating biased sensitivity for the three growth phases. In addition, the smallest $V_s$ was detected in Ile, suggesting equivalent sensitivity all growth phases. Taken together, Ile and/ or branched-chain AAs (BCAAs) participate commonly in all growth phases and are probably global coordinators for bacterial growth. This finding was independent of the methods used for the evaluation of the variance (*Figure 5—source data 1*).

## Risk diversification strategy for population survival

The three components serine, sulfate, and glucose, determining the growth lag, growth rate, and growth yield, respectively (*Figure 6A*), could be categorized into the three major elements of nitrogen (N), sulfur (S), and carbon (C). The contribution and mechanism of C and N to population dynamics have been intensively studied (*Brown et al., 2014*; *Côté et al., 2016*; *El Zahed and Brown, 2018*; *Egli, 1991*), whereas little is known concerning S. To link sulfate to growth, flux balance analysis (FBA) (*Orth et al., 2010*) simulation was performed. The result showed that a decreased growth rate was associated with an increased rate of sulfate uptake (*Figure 6B*), supporting the determinative contribution of S to the growth rate. Nevertheless, the FBA simulation did not provide a perfect explanation, as the concentration of sulfate used in the simulation was somehow excessive compared to that used for culture in general. The determinative role of S ($SO_4^{2-}$) in the growth rate might be related to its function as a material because it is not only a major constituent of the earth but also the major element in organisms (*Morgan and Anders, 1980*; *Heldal et al., 1985*; *Novoselov et al., 2013*). Since S ($SO_4^{2-}$) was a highly reactive chemical, e.g., exposure to $SO_4^{2-}$ increased reactive oxygen species levels in bacteria (*Chen et al., 2016*), its determinative role in growth rates was probably mediated by the stress response.

Notably, the different elements regulating various growth phases strongly implied risk diversification in fate decisions as a survival strategy. To demonstrate whether the differentiation of elements for growth decisions are a practicable survival strategy, theoretical simulations based on either a single or multiple determinants for the three parameters were additionally performed. Every 1000 simulations

**Figure 7.** Correlation of the growth parameters. (**A**) Density plots of the three parameters. Pairs of the three parameters lag time ($\tau$), growth rates ($r$), and saturated population size ($K$) are plotted as dots. The colour bars indicate the numbers of data points. Spearman's correlation coefficients and the p values are indicated. (**B**) Violin plots of the final population size. Relative population size of every 10,000 simulations considering the correlation coefficients of any pairs of the three parameters $\tau$, $r$, and $K$ is shown. Statistical significance of the Mann-Whitney U test is indicated.

were conducted at the varied threshold ($d$), i.e., the ratio was defined as population extinction. The results showed that a three-component set of decision makers led to a higher probability of survival, particularly when raising the extinction threshold (*Figure 6C*). It suggested that the differentiation in fate decision makers prevented the bacterial population from undergoing extinction more competently than the single decision maker did. It must be beneficial for the bacteria growing in a fluctuating environment, as it agreed well with the prospected Y-A-S strategy of the microorganisms in nature, i.e., the growth strategy for high yield, resource acquisition, and stress tolerance, respectively (*Malik et al., 2020*). In addition, the simulated result of reduced risk of extinction mediated by the differentiation in decision makers seemed to be biologically reasonable. The previous studies observed the trophic (*Sanders et al., 2018*) and functional (*Li et al., 2021*) redundancy in ecosystems and the genetic redundancy in living cells (*El-Brolosy and Stainier, 2017*), which demonstrated that the survivability was secured by the participation of multiple factors. It must be beneficial for maintaining the robustness of ecosystems and cells.

## Coordination in bacterial population dynamics

As the differentiation in decision-making components for bacterial growth allowed the independent decision for varied growth phases, the previously reported correlated changes in the growth parameters (*Novak et al., 2006*; *Engen and Saether, 2006*; *Basan et al., 2020*; *Liu et al., 2006*; *Nishimura et al., 2017*) were supposed to be weakened. However, the three parameters remained significantly correlated (*Figure 7A*), which indicated that the risk diversification strategy did not disturb the trade-off or coordination, e.g., $K/r$ selection (*Cavalier-Smith, 1980*). The correlations demonstrated that $\tau$, $r$, and $K$ were highly dependent, which well explained why the multimodal distributions of the growth parameters led to only four PCA clusters (*Figure 2*). Considering the dependency among the three growth parameters, which were decided by three different chemicals (*Figure 6A*), every 10,000 simulations of population dynamics considering the correlation coefficients (*Figure 7A*) were additionally performed. The results showed that the three decision makers facilitated the larger population size than the single decision maker did (*Figure 7B*), revealing that the differentiation in decision-making chemicals benefited the bacteria in maintain the final population size.

The correlated changes of the growth parameters might be due to the global participation of Ile and/or BCAAs, as the decision makers and common sensors. The frequency of Ile and BCAAs coded into the proteins in growing cells was evaluated according to the expression levels of the genes coding for the proteins (*Figure 8A*). The relative abundance of AAs was determined as the ratio of the frequency of the target AA to the sum of all 20 AAs in all proteins. Taking into account the variation in the copy number of proteins in growing cells, the frequency of each AA was normalized based on the relative abundance of gene expression (*Figure 8B*). The relative expression level of each gene (protein) was calculated as the mean of biologically repeated transcripts according to previous reports (*Liu et al., 2006*; *Ying et al., 2013*). The results showed that the relative abundances of intracellular Ile and BCAAs were significantly higher than their theoretical ratios, i.e., 1 or 3 out of 20 AAs, 5 or 15%, respectively (*Figure 8C*). Although the most abundant AA was not Ile but Leu (*Figure 8—figure supplement 1*), their regulation and metabolism are closely related (*Newman et al., 1992*). The results revealed that the protein building blocks required more BCAAs than other AAs, except Ala and

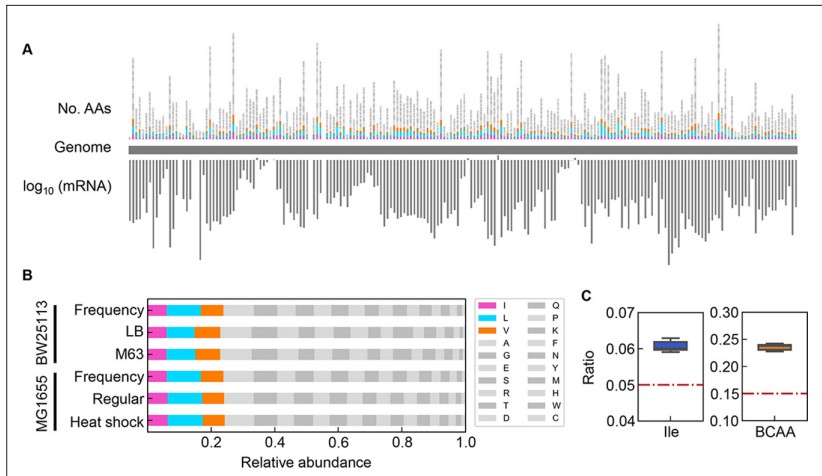

**Figure 8.** Relative abundance of branched-chain amino acids (BCAAs) in growing *E. coli*. (**A**) Chromosomal distribution of 20 amino acids (AAs). The numbers of 20 AAs coded into the proteins are indicated with the upward vertical bars at the chromosomal positions of the corresponding genes. BCAAs and the remaining 17 AAs are in colour and monotone, respectively. The expression levels of all genes coding for the proteins are shown in a logarithmic scale and are indicated by the downward vertical bar in grey. (**B**) Relative abundance of 20 AAs. Twenty AAs are shown with a single letter abbreviation. BCAAs are highlighted. The *E. coli* strains BW25113 and MG1655 are indicated. Frequency represents the relative abundance of the AAs, while all proteins encoded on the genome are of equivalent amount. LB, M63, regular and heat shock indicate the relative abundance of the AAs according to the transcriptomes of the *E. coli* cells grown in LB, in M63, at 37°C and at heat shock conditions, respectively. (**C**) Relative abundance of Ile and BCAAs in growing *E. coli*. The boxplots represent the relative ratios estimated according to the genome and transcriptome information, and the red lines indicate the theoretical ratios of the 20 AAs.

The online version of this article includes the following figure supplement(s) for figure 8:

**Figure supplement 1.** Relative abundance of 20 amino acids (AAs) in growing *Escherichia coli*.

Gly (*Figure 8—figure supplement 1*). The coordination among the three growth parameters might be balanced by BCAAs. In addition, the correlations of the growth parameters to each other and to the chemical gradients were detected at a population level (as shown in *Figure 4* and *Figure 7*), although both the cellular status and the environmental condition must have been fluctuated and changed along with the bacterial growth (i.e. batch culture). These coordinated alterations indicated the homeostasis in the complex systems, e.g., living cells and ecosystems.

## Discussion

The experimental restriction of the present survey was supposed to be the variety of environmental conditions and the interval of the concentration gradient. First, the study focused on the contribution of chemical conditions to bacterial growth, which was also affected by other environmental conditions, such as temperature and oxygen (*Ratkowsky et al., 1982*; *Baig and Hopton, 1969*; *McDaniel and Bailey, 1969*). As the quantitative adjustment of these conditions was currently impracticable for the high-throughput assay, these conditions were beyond the scope of the present study. The findings on the contribution of medium components to bacterial growth were true under laboratory conditions. Second, the concentration gradients of the components were prepared as broadly as possible to achieve the maximal solubility available for medium combinations, some of which were largely different from those used in the laboratory and/or found in nature (*Heldal et al., 1985*; *Novoselov et al., 2013*). The broad range of concentrations allowed us to acquire a boundless fitness landscape across the greatest environmental gradient and led to a wide concentration interval, i.e., changes on a logarithmic scale, in the growth assay. The concentration gradient of the most sensitive range to change bacterial growth might have been missed or masked. As the proper range of concentration gradients for sensitive growth change remained a black box, practicable conditions were applied. Theoretically, the issue concerning the best concentration gradient could be solved by extensive

growth assays associated with the ML prediction of concentration determination (rescale), i.e., introducing the predicted concentration to the following round of the growth assay and repeating the ML training and the experimental test. The extended repeats would result in the best medium combination for bacterial growth, which could be applied for culture optimization and development.

As the present study was to investigate the contribution of medium variation to bacterial population dynamics, the differentiation was only made in the media, and the cell populations (stocks) were strictly controlled to be equivalent. A total of 12,828 growth assays used the identical cell population cultured in the minimal medium. The additional pre-culture in individual medium combinations, which was often performed in microbiology studies, would change the initial state of the cell population (*Figure 2—figure supplement 2*). If 966 subcultures had been performed to 966 media independently, the study would become investigating the contributions of 966 different media to 966 different cell populations. The experimental tests verified that the repeated transfer caused the continuous growth improvement (*Figure 2—figure supplement 3*), as commonly reported in the experimental evolution (*Kurokawa et al., 2022*; *Barrick et al., 2009*). Since the initial cell population was grown in the minimal medium, which was different from 966 medium combinations, the growth evaluated here could be considered as the adaptiveness in response to the environmental changes. The present survey somehow discovered the fitness landscape of the bacterial cells across a wide and complex chemical space. In addition, the concentrations of medium components usually altered accompanied by the population increase in the growth assays. Such fluctuation in chemical concentrations was out of consideration in the analyses, which was a common limitation in batch cultures. Strictly speaking, the present study provided the dataset connecting the initial concentration of chemical components to the maximal growth rate that the cells could achieve. As the initial chemical concentration could be easily manipulated and precisely controlled, the results here were assumed to be applicable in preparing the medium for desired bacterial growth.

Since the accuracy and reliability of ML are largely dependent on the quality and quantity of the training data, the impact of the experimental data on the ML models was carefully assessed. First, the representative ML models and a commonly used statistic model of multiple regression were compared. Although multiple regression is known to have the highest interpretability, its accuracy of predictability was likely to be worse than that of the ML models (*Figure 3—figure supplement 1*). The results well supported the common sense that the ML approach was more suitable for studying the complex systems, which were the growing bacterial cells and the chemical media in the present survey. Additionally, among the tested ML models, the best accuracy was acquired with the ensemble model; nevertheless, as it required the longest time for model training (*Figure 3—figure supplement 2*) and was uninterpretable, the GBDT model was finally employed. Second, whether the abundance of the dataset affected the accuracy of the GBDT model was evaluated. The amount of data used for model training varied from 10 to 90% of the entire big dataset. Although a small amount of data (~10%) led to a relatively high accuracy on average, the variance in the accuracy of repeated model training was too large to reach a reliable prediction (*Figure 3—figure supplement 3*). An increase in the abundance of the training data decreased the variance of the model accuracy (*Figure 3—figure supplement 3*), demonstrating that a sufficiently large dataset was essentially required to achieve robust ML prediction for the biological experiments, as discussed in the different ML-associated microbial studies (*Topçuoğlu et al., 2020*). The dataset used here was large enough to grant a small variance, indicating the robust result of model training. Third, whether the accuracy of prediction was attributed to the experimental errors was evaluated. An equal amount of training data (n=400) with varied experimental accuracy, i.e., the variance of biological replication (CV = 0.05–0.12), was used to test the accuracy of the ML models. Intriguingly, the training data with large variance, i.e., a large experimental error caused by biological replication, resulted in the high accuracy of ML in comparison to those with small variance, which led to the decreased accuracy of ML (*Figure 3—figure supplement 4*). Accordingly, the entire experimental dataset, regardless of the experimental error, was used for ML to draw the conclusion presented here.

The multimodal distributions of the three parameters reflected a broad variation in medium combinations. Whether the main conclusion regarding the differentiated decision makers for varied growth phases was biased by the medium combinations was also evaluated. First, the variation in the concentration of each component was counted. The most abundant variation of concentration was that for the chlorine ion (Cl), which was a low-priority contributor to growth, whereas the decision-making

components showed either high or low variation of concentrations, such as for sulfate or glucose, respectively (*Figure 1—figure supplement 1*). Second, although the AAs presented equivalent variations in concentration, only Ile, Ser, and Leu were determined to be the growth determinative components. Finally, even if the multimodal distributions of the three parameters were arbitrarily divided into two monomodal-like distributions for data separation, which led to the reduced abundancy of the dataset, the differentiation in decision-making components for the three growth phases remained (*Figure 4—figure supplements 2–4*). As the data separation reduced the variety of medium combinations, the highest-priority components were either similar or varied from those identified while using the whole dataset. This result indicated that the diversity of experimental conditions, i.e., the abundance of training data, could influence the ML prediction. The present study applied the exceeding range of the concentration gradient and the high variability of medium combinations, which might cover the landscape of population dynamics as broadly as possible in the laboratory; therefore, the finding of the differentiation in the components deciding the three growth phases was independent of the experimental restriction.

In summary, the present study provided an informative and quantitative big dataset relating bacterial growth (population dynamics) to environmental factors as a successful example of a combination of high-throughput data generation and ML. Using a simple ML model to evaluate three growth parameters was likely sufficient to capture the bacterial population dynamics in well-controlled conditions. The differentiation in decision-making components for the lag, exponential, and stationary phases protected the bacterial population against extinction. This finding revealed a common and simple strategy of risk diversification for bacterial growth in conditions of excessive resources or starvation, which is a reasonable approach in evolution and ecology. As a representative demonstration, this study showed that investigating the microbial world by data-driven approaches allows us to perceive highly intriguing insights that were inconceivable by traditional biological experiments. Nevertheless, the ML-assisted approach remains as an emerging technology and is required to improve its biological reliability and accessibility for common applications in the studies of life science, microbiology, and ecology.

## Materials and methods
### Bacterial strain and stock preparation
The wild-type *E. coli* strain BW25113 was used, which was provided by the National BioResource Project (National Institute of Genetics, Shizuoka, Japan). To reduce the experimental errors of the repeated growth assay on different days, common stocks of the exponentially growing *E. coli* cell culture were prepared beforehand, as described previously (*Kurokawa and Ying, 2017*). In brief, the *E. coli* cells were cultured in 5 mL of M63 minimal medium using a bioshaker (BR23-PF, Taitec) at 200 rpm and 37°C. The cell culture was stopped when its optical density measured at 600 nm ($OD_{600}$) reached ~0.1. The culture was subsequently divided into a small portion (60 µL) in 1.5 mL microtubes (Watson) and stored at –80°C for future use. Hundreds of aliquots (stocks) were prepared at once and disposably used in the growth assay; that is, the aliquots were used only once, and remaining cultures were discarded.

### Medium composition and combinations
A total of 44 pure chemical substances, determined according to the literature (*Oberhardt et al., 2015*; *Neidhardt et al., 1974*), were all commercially available (Wako or Sigma). The minimal concentrations of these compounds were set at zero in general, and the maximal concentrations were determined individually according to the literature or laboratory manuals. In addition, the concentrations of the compounds rarely used in the known media were experimentally examined (*Figure 1—figure supplements 2 and 3*). According to the determined maximal concentration, stock solutions of these chemical substances were prepared in advance for the easy preparation of medium combinations. The chemical substrates were dissolved in highly pure water (Direct-Q UV, Merck) at high concentrations. Subsequently, the resultant solutions were sterilized, either using a sterile syringe filter with a 0.22 µm pore size and hydrophilic PVDF membrane (Merck) for those heat sensitive compounds or by autoclaving at 121°C for 20 min. The stock solutions were divided into aliquots (10–100 µL) in 1.5 mL microtubes (Watson) and stored at –30°C for future use. A total of 100–300 stocks were prepared at

once for individual chemical substrates. To avoid repeated thawing and freezing of the stock solutions, aliquots were used only once. The medium combinations were prepared by mixing the stock solutions (aliquots) just before the growth assay. The concentrations of the substrates were varied on a logarithmic scale, and only a single substrate was altered for each assay. A total of 966 combinations were tested in the growth assay (*Figure 2—source data 1*).

## Growth assay

The high-throughput growth assay was conducted to acquire the growth curves in the medium combinations, as described previously (*Ashino et al., 2019*). The culture stocks were diluted 1000-fold with 5 mL of fresh media of varying medium combinations in 5 mL tubes (Watson). The diluted cell culture mixtures were loaded into a 96-well microplate (Costar) in four-to-six wells (200 μL per well) with varied locations per medium combination. The 96-well plates were incubated in a plate reader (Epoch2, BioTek) with a rotation rate of 567 rpm at 37°C. The temporal growth of the *E. coli* cells was detected at an absorbance of 600 nm, and the readings were obtained at 30 min intervals for 24–48 hr. A total of 12,828 reliable growth curves were acquired.

## Data processing and calculation of the growth parameters

The temporal $OD_{600}$ reads were exported from the plate reader and processed with Python, as described in detail elsewhere (*Ashino et al., 2019*). The growth parameters $\tau$, $r$, and $K$ were evaluated according to previous reports (*Ashino et al., 2019*; *Kurokawa et al., 2016*) using a previously developed Python program (*Ashino et al., 2019*). In brief, $\tau$ was determined as the time when the increase in $OD_{600}$ was observed in five consecutive reads; $r$ was defined as the mean of three continuous logarithmic slopes of every two neighbouring $OD_{600}$ values within the exponential growth phase using 'gradient' in the 'numpy' library; and $K$ was calculated as the mean of three continuous $OD_{600}$ values including the maximum, which was determined using 'argmax' in the 'numpy' library.

## PCA and clustering

PCA (*Ringnér, 2008*; *Abdi and Williams, 2010*) was performed using 'PCA' in the 'decomposition' module from 'scikit-learn' (*Pedregosa, 2011*). The concentrations of 41 components were normalized within one unit, and the 966 combinations were used as input. The principal component scores of PC1 and PC2 were used for the correlation analysis of the three growth parameters. Clustering of the PC1–PC2 scores was performed using 'KMeans' in the 'cluster' module of the 'scikit-learn' library.

## ML models and multiple regression model and evaluation

ML was performed using a supercomputer, the Cygnus system (NEC LX 124Rh-4G). The ML models of GBDT, k-nearest neighbour (k-NN), neural network (NN), random forest, support vector machine (SVM), and multiple regression were performed using 'GradientBoostingRegressor' in the 'ensemble' module, 'KNeighborsRegressor' in the 'neighbors' module, 'MLPRegressor' in the 'neural_network' module, 'RandomForestRegressor' in the 'ensemble' module, 'SVR' in the 'svm' module, and "LinearRegression" in the 'linear_model' module, respectively. The ensemble model was performed using 'StackingRegressor' in the 'ensemble' module and 'LinearRegression' in the 'linear_model' module. Data normalization was performed using 'StandardScaler' in the 'preprocessing' module for k-NN, NN and multiple regression, and 'MinMaxScaler' in the 'svm' module for SVM. All these modules were in the 'scikit-learn' library.

A fivefold nested cross validation was performed to evaluate the ML models. A grid search was used for the hyperparameter search, as follows. In the GBDT model, 'random_state' and 'n_estimators' were configured as 0 and 300, respectively; 'learning_rate' and 'max_depth' were searched from 0.001 to 0.5 in increments of 0.005 and among 2, 3, 4, and 5, respectively. In the k-NN model, 'n_neighbors' was searched among 1, 2, 3, and 4. In the NN model, 'solver' and 'alpha' were configured as 'adam' and 0.001, respectively; 'hidden_layer_sizes' was searched among (100,100,100), (100,100), (50,50), and (50,50,50). In the random forest model, 'random_state' and 'n_estimators' were configured as 0 and 300, respectively; 'max_depth' was searched among 2, 3, and 4. In the SVM model, the 'kernel' was configured as 'rbf'; 'C', 'gamma', and 'epsilon' were searched from $2^{-5}$ to $2^{10}$, $2^{-20}$ to $2^{10}$, and $2^{-10}$ to $2^{0}$, respectively, in increments of $2^{2}$. All other hyperparameters were used as default. A fivefold cross validation was performed to evaluate the multiple regression model.

The metrics adopted to estimate the accuracy of the ML models were determined as follows. The coefficient of determination ($R^2$), mean squared error (MSE), mean absolute error, and explained variance score were calculated using 'r2_score', 'mean_squared_error', 'mean_absolute_error', and 'explained_variance_score' in the 'metrics' module of the 'scikit-learn' library, respectively. The root mean squared error was calculated with the MSE values using 'sqrt' in the 'numpy' library. For each metric, Scheffe's multiple comparison procedure was used to test for differences in the ML and multiple regression models.

## GBDT prediction

A regression model was created by using the log-transformed concentrations of the components. The 'feature_importances_' attribute represents the importance of each component to the creation of the model. Outer and inner cross validation was performed using 'cross_val_score' in the 'model_selection' module of the 'scikit-learn' library. The hyperparameters were searched using 'GridSearchCV' in the 'model_selection' module of the 'scikit-learn' library. 'learning_rate' and 'max_depth' were searched from 0.01 to 0.5 in increments of 0.01 and among 2, 3, 4, and 5, respectively. 'n_estimatiors' was configured at 300, and the other hyperparameters were set to default values. The 'feature_inportance_' values were calculated by fivefold cross validation, and the mean of the five values was used as the result of the GBDT prediction.

## Evaluation of sensitivity

The changes in the growth parameters associated with the concentration gradient of each component were evaluated by curve fitting of a cubic polynomial as described previously (**Kurokawa et al., 2021**).

$$S_{p,i} = 1 - \left( Area \times p_{i,max}^{-1} \times \left( x_{max} - x_{min} \right)^{-1} \right) \tag{1}$$

Here, $S_{p,i}$, $Area$, $p_{i,max}$, $x_{min}$, and $x_{max}$ represent the sensitivity evaluated with any of the growth parameters in condition $i$, the area under the regression curve, the largest value of the growth parameter in condition $i$, and the minimum and maximum concentrations of each chemical component, respectively. The sensitivity was further evaluated as follows.

$$S_p = \frac{1}{n} \times \sum_{i=1}^{n} S_{p,i} \qquad (n = 6, p = \tau, r, K) \tag{2}$$

$$S_g = S_\tau + S_r + S_K \tag{3}$$

$$V_s = \sqrt{\frac{\sum \left( \frac{S_p}{S_g} - \bar{S} \right)^2}{m-1}} \qquad (m = 3) \tag{4.1}$$

$$\bar{S} = \frac{1}{3} \times \frac{S_\tau + S_r + S_K}{S_g} \tag{4.2}$$

Here, $S_\tau$, $S_r$, and $S_K$ represent the sensitivity of $\tau$, $r$, and $K$, respectively. $S_g$ and $V_s$ represent the sum and the variance of $S_\tau$, $S_r$, and $S_K$, respectively. Additionally, four different methods were applied to estimate $V_s$, as follows.

$$Crit. 1 = \frac{S_\tau + S_r + S_K}{S_g} \tag{5}$$

$$Crit. 2 = \frac{S_\tau + S_r + S_K - 3S'}{S_g} \tag{6}$$

$$Crit. 3 = \sqrt{\frac{\sum (S_p - s')^2}{m-1}} \tag{7}$$

$$Crit. 4 = min \left( \angle S_\tau, \angle S_r, \angle S_K \right) \tag{8}$$

Here, $S'$ indicates the mean of $S_\tau$, $S_r$, and $S_K$. $\angle S_\tau$, $\angle S_r$, and $\angle S_K$ represent the three angles calculated from the triangle (**Figure 5E**).

## FBA simulation

FBA simulation was performed using the open software COBRAme (**Lloyd et al., 2018**) iJL1678b-ME and qMINOS, which were available in the Docker images (**Lloyd et al., 2018**), were used as the model and the solver, respectively, where 'mumax' and 'precision' were set as 2 and 1E-6, respectively. 4

out of 41 components, i.e., VB9, VB2, borate, and PABA, were excluded in the simulation, as they were absent in the ME model. The lower bounds of the efflux of the components were set as the negative values, which allowed the *E. coli* cells to take them up from the media. The lower bound of the efflux of AAs and citrate was set to –10, and –1000 was set for the others. The lower bounds of the efflux of selenite, selenite, tungstate, Li, Sc, and Tl were set to zero, and those of cobalt, Mn, Ni, RNase_m5, RNase_16, and RNase_m23 were set as −0.00001, −0.001, −0.001, −1, −1, and −1, respectively, because these components were absent in the present study. The uptake of sulfate was fixed by setting the upper bound of the efflux to a negative value to predict the growth rate when the uptake of sulfate was varied.

## Genomic datasets and annotation

The genome and transcriptome datasets of the *E. coli* BW25113 and MG1655 strains were obtained from GenBank (CP009273 and NC_000913) and GEO (GSE33212 and GSE136101), respectively. The gene (protein) annotation and counting of the AAs were processed using BioPython (***Cock et al., 2009***).

## Theoretical simulation of survival probability

The simulation of population dynamics over 24 hr was conducted according to the following equations.

$$N_0 = K_{max} \times 0.001 \tag{9}$$

$$\tau = T \times \tau_r \ (T = 24) \tag{10.1}$$

$$r = r_r \tag{10.2}$$

$$K = K_{max} \times K_r \ (K_{max} = 1) \tag{10.3}$$

Here, $N_0$, $K_{max}$, $\tau_r$, $r_r$, and $K_r$ are the initial population, the population maximum, and the three variables $\tau$, $r$, and $K$, respectively. In the case of triple independent decision makers, the values of $\tau_r$, $r_r$, and $K_r$ were randomly selected from 0 to 1 without coordinated change. In the case of a single common decision maker, once any of the three parameters was randomly selected from 0 to 1, the other two were decided as follows.

$$\tau_r = 1 - r_r \tag{10.4}$$

$$K_r = 1 - r_r \tag{10.5}$$

The population dynamics were defined as follows.

$$t_i < \tau \ , \ N(t_i) = N_0 \tag{11.1}$$

$$t_i \geq \tau \ N(t_i) = \frac{K + N_0}{N_0 + (K - N_0)e^{-rt_j}} \tag{11.2}$$

$$t_j = t_i - \tau \tag{11.3}$$

where $N(t_i)$, $t_j$, and $t_i$ are the population size at time $t_i$, any time point within the exponential phase, and any time point from 0 to 24 hr in a 0.5 hr interval, respectively. Whether the population was extinct or survived was determined according to the survival threshold, $d$, as follows.

$$N(24) < d, \text{death} \tag{12.1}$$

$$N(24) \geq d, \text{survival} \tag{12.2}$$

$$d = K_{max} \times d_r \tag{12.3}$$

Here, $N(24)$ and $d_r$ are the final population size at 24 hr and the threshold varying from 0 to 1 in increments of 0.05, respectively. The survival probability was defined as the frequency of 'survival' in every 1000 simulations at each $d_r$.

## Theoretical simulation of population dynamics considering the correlation

Considering the correlations among the growth parameters, $\tau_r$ in 10.1 (denoted $\tau_{r\_Tcorr}$) was randomly varying from 0 to 1, and $r_r$ in 10.2 (denoted as $r_{r\_Tcorr}$) and $K_r$ in 10. 3 (denoted as $K_{r\_Tcorr}$) were determined as follows.

$$r_{r\_Tcorr} \;=\; C_{\tau r}\left(1 - \tau_{r\_Tcorr}\right) + \left(1 - C_{\tau r}\right) I_r \tag{13.1}$$

$$K_{r\_Tcorr} \;=\; C_{\tau K} \times \tau_{r\_Tcorr} + C_{rK}\left(1 - r_{r\_Tcorr}\right) + \left(1 - C_{\tau K} - C_{rK}\right) I_K \tag{13.2}$$

Here, $C_{\tau r}$, $C_{\tau K}$, and $C_{rK}$ represented the correlation coefficients of any pairs of $\tau$, $r$, and $K$, respectively. According to the correlation coefficients acquired from the growth assay (**Figure 7A**), $C_{\tau r}$, $C_{\tau K}$, and $C_{rK}$ were set to 0.74, 0.58, and 0.17, respectively. In addition, $I_r$ and $I_K$ were variables randomly selected from 0 to 1. Simulation of population dynamics was performed according to 10.3 and 11.3, and the relative population size at 24 hr, i.e., $N(24)$, was calculated consequently for 10,000 times.

## Separation of the multimodal distributions

Gaussian kernel density estimation was used to determine the boundaries of the multimodal distributions, which were considered as bimodal, for data separation of the growth parameters. The probability density function was conducted using 'gaussian_kde' in the 'stats' module of the 'scipy' library, in which 'bw_method' was configured as 0.3. These distributions were divided vertically into 1000 equal areas. The trough point, i.e., the smallest area, for data separation was determined using 'argrelmin' in the 'signal' module of the 'scipy' library. The three growth parameters were independently divided into two datasets of low and high mean values. The following GBDT prediction of $\tau$, $r$, and $K$ was performed separately.

## Acknowledgements

We thank NBRP (Japan) for providing the *E. coli* strain. This work was supported by a JSPS KAKENHI Grant-in-Aid for Challenging Exploratory Research (grant number 21K19815) and partially by a JSPS KAKENHI Grant-in-Aid for Scientific Research (B) (grant number 19H03215).

# Additional information

### Competing interests

Honoka Aida, Takamasa Hashizume: The medium combinations were submitted for a patent under the control number of 2021-171528 (Japan). The other authors declare that no competing interests exist.

### Funding

| Funder | Grant reference number | Author |
| --- | --- | --- |
| Japan Society for the Promotion of Science | 21K19815 | Bei-Wen Ying |
| Japan Society for the Promotion of Science | 19H03215 | Bei-Wen Ying |

The funders had no role in study design, data collection and interpretation, or the decision to submit the work for publication.

### Author contributions

Honoka Aida, Resources, Data curation, Software, Formal analysis, Validation, Visualization, Methodology, Writing – original draft; Takamasa Hashizume, Software, Formal analysis, Methodology; Kazuha Ashino, Resources, Methodology; Bei-Wen Ying, Conceptualization, Resources, Data curation, Formal analysis, Supervision, Funding acquisition, Validation, Investigation, Visualization, Methodology, Writing – original draft, Project administration, Writing - review and editing

### Author ORCIDs
Bei-Wen Ying (iD) http://orcid.org/0000-0003-2517-5686

### Decision letter and Author response
Decision letter https://doi.org/10.7554/eLife.76846.sa1
Author response https://doi.org/10.7554/eLife.76846.sa2

## Additional files

### Supplementary files
• Transparent reporting form

### Data availability
All data generated or analysed during this study are included in the manuscript and supporting file.

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
