## [Editor Report]

In this manuscript, the authors quantitatively analyze the growth curves for *E. coli* under a large number of growth conditions and use different machine learning methods to tackle the combinatorial complexity of conditions as well as to predict growth parameters from media composition. The large datasets and the use of ML to handle such complex modeling will be of general interest to the biology community.

---

## [Decision Letter]

**Decision letter after peer review:**

Thank you for submitting your article "Machine learning-assisted discovery of growth decision elements by relating bacterial population dynamics to environmental diversity" for consideration by *eLife*. Your article has been reviewed by 3 peer reviewers, one of whom is a member of our Board of Reviewing Editors, and the evaluation has been overseen by Naama Barkai as the Senior Editor. The following individual involved in the review of your submission has agreed to reveal their identity: Anne Thessen (Reviewer #2).

Essential revisions:

1) There are concerns about the robustness of the experimental results e.g. because cells were not allowed to adapt and because consumption of nutrients was not accounted for. While redoing all the experiments would be too cumbersome, we would recommend that a few confirmatory experiments (e.g. comparing growth curves for adapted and non-adapted strains) be performed.

2) The strength of the machine learning approach (compared to simpler and easier to interpret methods such as principal component analysis, partial least square regression, etc.) needs to be documented. Please introduce a metric that helps the reader better assess the strength of the statistical analysis

3) A stronger discussion of the approach and/or analysis and/or results will be necessary to highlight the novelty of this study. In its present version, reviewers read underwhelmed by the findings.

*Reviewer #1 (Recommendations for the authors):*

1) Panels in Fig. 4B would benefit from a violin representation in order to better represent the distribution of parameters. The concentration dependence for the lag time and the growth rates do not look graphically striking while the cor and p values look impressive. The authors should comment on this discrepancy.

*Reviewer #2 (Recommendations for the authors):*

I have two methodological concerns.

1. Cells adapting to growth media: First test to see if you have this problem by finding out what growth media was used at the culture facility where this strain was acquired. Do your data show any correlations with the conditions similar to that growth media and a short lag phase, high max growth rate, or high max biomass? Is this strain growing better in growth media that more closely resembles the growth media it is already used to? The best solution is to go back and do the experiments again with adapted cultures, but I know that is too much work.

2. Changing chemical concentrations and ratios: The easy fix is to acknowledge and discuss this as a limiting factor in your conclusions or explain how this does not matter for this particular study. The best you can say is that the max growth rate is highest in growth media with high initial sulphate concentrations (or some equivalent statement). Sometimes the ratios of chemicals can be more important than the concentration. You can also be seeing the effects of different metabolites being released into the growth media.

Alternatively, you could make this paper about ML methods and replace the biology-specific claims with a very small literature search that shows your results are biologically plausible.

Keep in mind that even though you are doing a highly controlled experiment, the cells are adapting, the population is evolving, and the amounts and types of chemicals in the media are changing.

Specific Issues

1. I don't see that the data are available.

2. Is the gene expression in Figure 8 actually "differential" gene expression? If so, what is it compared to?

3. Considering the problems with the biology, I'm not convinced you can predict the best growth media for a desired outcome.

4. Please use active voice.

5. Check your results to see if osmotic balance is having any effect (since you are using "highly pure water").

6. Autoclaving can cause media to change in important ways. Make sure you did not have this problem.

7. There are results from simulations present, but no discussion of if they are biologically plausible.

8. I don't think you should be trying to do a prediction task at this point (S3).

*Reviewer #3 (Recommendations for the authors):*

1) I can't find what the paper is solving that to date was not known. Why should the community care or take home?

2) PCA plot and text show that there is an association between media components and *E. coli* growth. Isn't that obvious?

3) 6 different off-shelf ML models are used for the regression and one was chosen for interpretation. That's fine but I do not see the innovation here. It say like to say doing PCR to amplify a gene or CRISPR for a deletion. Off-the-shelf ML methods are now established protocols to run classification/regression when the predictor matrix is very big. What is new here? The application to *E. coli* growth curve inferred parameters?

4) Growth decision-making. Is this term established and defined in the field. Here it is not explained. Also, who is deciding, the bacteria? Bacteria do not decide.

5) Sensitivity analysis section is not clear and I especially do not understand how the authors arrive at the final claim (sentence) in that section.

6) Line 165-167. I do not understand the sentence. Are the authors claiming S (sulphur) to be the most abundant material on earth? What is the relationship with the results of the FBA modeling? I do not find a connection or at least it is not well explained.

7) The paper lacks validation. External validation. The authors make claims about bacteria being able to avoid extinction etc. Prove it. The authors are using *E. coli* growth curves one of the easiest and quickest experimental systems available. Would be great if they could experimentally validate what they claim. In this shape, it feels a bit of an elaborate way to analyse growth curve data.

---

## [Author Response]

Essential revisions:Reviewer #1 (Recommendations for the authors):1) Panels in Fig. 4B would benefit from a violin representation in order to better represent the distribution of parameters. The concentration dependence for the lag time and the growth rates do not look graphically striking while the cor and p values look impressive. The authors should comment on this discrepancy.

We do agree that the violin plot well illustrates the data distribution and is to identify the difference among the varied concentrations as a qualitative viewpoint. Here, the dot scatter plot was used to match the correlation analysis, which presents the relationship of two parameters (i.e., the chemical concentration and the growth parameter) as a quantitative viewpoint. For reference, the violin plots were additionally made and newly supplied in the revision (Figure 4−figure supplement 1).

The concentration dependence of τ and r seemed to be not prominent graphically might be due to the large size of the data set. The larger number of data usually resulted in the higher significance, as the statistical significance was evaluated according to the Spearman's rank correlation. The discrepancy between the graphics and the statistics was additionally discussed as follows (lines 123~125).

“Since the Spearman’s rank correlation was used, the large size of dataset led to high significance, which was somehow discrepant to the graphics.”

Reviewer #2 (Recommendations for the authors):I have two methodological concerns.1. Cells adapting to growth media: First test to see if you have this problem by finding out what growth media was used at the culture facility where this strain was acquired. Do your data show any correlations with the conditions similar to that growth media and a short lag phase, high max growth rate, or high max biomass? Is this strain growing better in growth media that more closely resembles the growth media it is already used to? The best solution is to go back and do the experiments again with adapted cultures, but I know that is too much work.

Thank you for your careful thinking. We do understand your concern. The critical question is how to make sure that the cells are adapted? Adaptation is a familiar term for us, as we have fruitful research experiences on experimental evolution. It’s well known in the field that the cells usually grow better/faster after the transfer/subculture in a new medium. What we are wondering is how many times of transfer (subculture) is enough for the cells to achieve the adaptive state. We do know that the overnight preculture (e.g., a single transfer, subculture) is commonly performed in biology studies, which is explained as the adaptation of the cells to the medium in the field of microbiology for decades. In fact, it’s largely uncertain whether the cells are truly adapted to the medium, following the subculture. To demonstrate what we are discussing here, the subculture (daily transfer) was newly performed with the same *E. coli* strain used in the present study. As described in the Materials and methods, the *E. coli* cells (stocks) were prepared with the preculture in the minimal medium (M63). The subculture was started with the cell stock identical to those used for 12,828 growth assays, and the daily transfer was performed in the same medium, which was one of 966 medium combinations. The results showed that the repeated transfers (subcultures) slightly increased the growth rates (Figure 2−figure supplement 3), although the subculture was initiated from the pre-cultured (pre-adapted) cell population. It demonstrated that the subcultures did not promise the adaptation of the cells to the media.

A total of 966 growth rates represents the adaptiveness of the identical cell population in 966 different media. As you mentioned that the cells are changing during growth, the subculture would cause the changes in cell population. The fitness landscape was illustrated as an example (Figure 2−figure supplement 2). The pre-culture changes the initial cell state, which leads to the cells of different initial states (differentiation in subcultures) to climb the different fitness mountains.

If 966 subcultures had been performed to 966 media independently, then the study investigated the contributions of 966 different media to 966 different cell populations. As our study is to investigate the contribution of medium variation to cell growth, the changes are expected to be happened to the medium but not to the cell population. This is the reason why 12,828 growth assays were under strict control to use the identical cell population without pre-culture here.

Taking account of this essential viewpoint, we added the interpretation and the supplementary figures in the Discussion as follows (lines 264~278).

“As the present study was to investigate the contribution of medium variation to bacterial population dynamics, the differentiation was only made in the media and the cell populations (stocks) were strictly controlled to be equivalent. A total of 12,828 growth assays used the identical cell population cultured in the minimal medium. The additional pre-culture in individual medium combinations, which was often performed in microbiology studies, would change the initial state of the cell population (Figure 2−figure supplement 2). If 966 subcultures had been performed to 966 media independently, the study would become investigating the contributions of 966 different media to 966 different cell populations. The experimental tests verified that the repeated transfer caused the continuous growth improvement (Figure 2−figure supplement 3), as commonly reported in the experimental evolution ^65,66^. Since the initial cell population was grown in the minimal medium, which was different from 966 medium combinations, the growth evaluated here could be considered as the adaptiveness in response to the environmental changes. The present survey somehow discovered the fitness landscape of the bacterial cells across a wide and complex chemical space.”

2. Changing chemical concentrations and ratios: The easy fix is to acknowledge and discuss this as a limiting factor in your conclusions or explain how this does not matter for this particular study. The best you can say is that the max growth rate is highest in growth media with high initial sulphate concentrations (or some equivalent statement). Sometimes the ratios of chemicals can be more important than the concentration. You can also be seeing the effects of different metabolites being released into the growth media.

Thank you for the crucial comment and the helpful advice. We do agree that the chemical concentrations were all indicated at t=0, as it’s unavailable to trace the changes of all chemicals in the medium during cell growth, particularly, in a high-throughput manner. The changes in chemical concentration must have been occurred, which was out of consideration in the present study. The benefit of using the initial concentration is the easy application, connecting the initial chemical concentration to the following population dynamics could allow us to predict cell growth from tested medium conditions. Here, we would like to adopt your idea of discussing it as a limiting factor as follows (lines 278~286).

“In addition, the concentrations of medium components usually altered accompanied by the population increase in the growth assays. Such fluctuation in chemical concentrations was out of consideration in the analyses, which was a common limitation in batch cultures. Strictly speaking, the present study provided the dataset connecting the initial concentration of chemical components to the maximal growth rate that the cells could achieved. As the initial chemical concentration could be easily manipulated and precisely controlled, the results here were assumed to be applicable in preparing the medium for desired bacterial growth.”

Alternatively, you could make this paper about ML methods and replace the biology-specific claims with a very small literature search that shows your results are biologically plausible.

The ML approach is somehow unfavored at the moment, as it doesn’t discover any specific molecular mechanism of clear biological function, which is often anticipated by researchers in biology studies. Alternatively, it provides global insights and laws on the biological systems and raises novel questions different from those raised in the experimental bioscience. We hope the present study could be accepted as an example of introducing ML to microbiology, which is supposed to be helpful for the future innovation in biology studies. Taking account of this viewpoint, the following sentence was added in the Discussion (lines 349~352).

“Nevertheless, the ML-assisted approach remains as an emerging technology and is required to improve its biological reliability and accessibility for common applications in the studies of life science, microbiology and ecology.”

Keep in mind that even though you are doing a highly controlled experiment, the cells are adapting, the population is evolving, and the amounts and types of chemicals in the media are changing.

It’s pleasant that we have the same thinking. Both the growing cells and the environmental chemicals are changing during the cultivation. Despite of the dynamically changing cells and environments, the correlations identified in the growth parameters and the medium combinations revealed somehow the robustness of living systems and the coordination in ecosystems. To emphasize this common sense/feature, the following sentences were added (lines 234~239).

“In addition, the correlations of the growth parameters to each other and to the chemical gradients were detected at a population level (as shown in Figure 4 and Figure 7), although both the cellular status and the environmental condition must have been fluctuated and changed along with the bacterial growth (i.e., batch culture). These coordinated alterations indicated the homeostasis in the complex systems, e.g., living cells and ecosystems.”

Specific Issues1. I don't see that the data are available.

The source datasets were associated with the main figures in the revision.

2. Is the gene expression in Figure 8 actually "differential" gene expression? If so, what is it compared to?

Sorry for the unclear description. Figure 8 showed the expression of all genes but not the differential gene expression. The genes coding for the proteins on the genome were indicated in Figure 8A, and the expression levels of these genes were multiplied by the number of the target amino acid in its coding protein to calculate the ratio of the amino acid (Figure 8B). The revision was made in both the Results (lines 219~221) and the figure legend (lines 822~823) as follows.

“The frequency of Ile and BCAAs coded into the proteins in growing cells was evaluated, according to the expression levels of the genes coding for the proteins (Figure 8A).”

“The expression levels of all genes coding for the proteins are shown in a logarithmic scale and are indicated by the downward vertical bar in grey.”

3. Considering the problems with the biology, I'm not convinced you can predict the best growth media for a desired outcome.

The purpose of the present study is not to predict the best medium, but to discover the growth law by connecting medium combinations to population dynamics. We believe that the ML-assisted approach must be valuable and useful for microbiology studies, although the methodology remains to be developed to improve its biological reliability. Actually, our ongoing studies using the present methodology successfully predicted/optimized the growth media for desired purposes, e.g., metabolite/substrate productivity (Uchida and Aida et al., in preparation). Considering this viewpoint, the following sentence was added in the Discussion (lines 349~352).

“Nevertheless, the ML-assisted approach remains as an emerging technology and is required to improve its biological reliability and accessibility for common applications in the studies of life science, microbiology and ecology.”

4. Please use active voice.

Thanks for the advice. The passive voice in writing is likely due to the old-fashioned language education. It’s true that the active voice is straightforward and easy understanding. We hope we are allowed to do it in our next paper.

5. Check your results to see if osmotic balance is having any effect (since you are using "highly pure water").

It’s true that the osmotic balance influences the cell growth. The variation in medium combination/composition causes the variation in osmotic balance. As the osmotic balance is resulted from the chemical combination/composition, the effect of osmotic balance is supposed to be represented by the contribution of medium combination/composition. If the study is to evaluate the contribution of a specific chemical, e.g., glucose, then the osmotic balance should be controlled as equivalent among the different test conditions. Since the present study is to evaluate the contribution of chemical combination, controlling the osmotic balance of all medium combinations to be the same is nonessential and unavailable. Relatively high contribution of Na and K to the cell growth (Figure 4A) indicated such osmotic effect. Taking account of the concern, the revision was made as follows (lines 116~121).

“The top ten features (i.e., components) contributing to the three parameters somehow overlapped (e.g., K, Na and phosphate), which might reflect the common effect of osmotic balance resulting from these components. Nevertheless, the components of the highest priority in governing the three parameters were highly differentiated, that is, serine, sulfate and glucose for *τ*, *r* and *K*, respectively (Figure 4A).”

6. Autoclaving can cause media to change in important ways. Make sure you did not have this problem.

Yes, we do understand that some compounds could be heat sensitive, such as vitamins and sugar. They were filter sterilized, as described in the Materials and methods as follows (lines 377-380).

“Subsequently, the resultant solutions were sterilized, either using a sterile syringe filter with a 0.22 µm pore size and hydrophilic PVDF membrane (Merck) for those heat sensitive compounds or by autoclaving at 121 °C for 20 min.”

7. There are results from simulations present, but no discussion of if they are biologically plausible.

Thank you for the comment. The previous studies reporting either ecological or genetic redundancy revealed the biological plausibility of the simulated result. Accordingly, the corresponding paragraph was revised associated with the additional references, as follows (lines 189~201).

“The results showed that a three-component set of decision-makers led to a higher probability of survival, particularly when raising the extinction threshold (Figure 6C). It suggested that the differentiation in fate decision-makers prevented the bacterial population from undergoing extinction more competently than the single decision-maker did. It must be beneficial for the bacteria growing in a fluctuating environment, as it agreed well with the prospected Y-A-S strategy of the microorganisms in nature, i.e., the growth strategy for high yield, resource acquisition and stress tolerance, respectively^55^. In addition, the simulated result of reduced risk of extinction mediated by the differentiation in decision-makers seemed to be biologically reasonable. The previous studies observed the trophic^56^ and functional^57^ redundancy in ecosystems and the genetic redundancy in living cells^58^, which demonstrated that the survivability was secured by the participation of multiple factors. It must be beneficial for maintaining the robustness of ecosystems and cells.”

8. I don't think you should be trying to do a prediction task at this point (S3).

Thank you for the suggestion. The figure and related description were deleted.

Reviewer #3 (Recommendations for the authors):1) I can't find what the paper is solving that to date was not known. Why should the community care or take home?

Sorry for the unclear writing. We do agree that this study is not to solve any specific problem being in the well-established biology communities/fields. The present study is to discover the novel biological features and/or laws in common by introducing the data-driven approaches to biological experiments. It shows a successful example of a combination of high-throughput data generation and machine learning, which should be valuable for the innovation and development in the communities/fields. Using a simple ML model to evaluate three growth parameters is likely enough to capture the bacterial population dynamics in well-controlled conditions. The novel finding of the differentiation in decision-making chemicals for bacterial growth allows us to propose a risk diversification strategy, which might be commonly adopted by microbes to survive in nature. To make the take-home message clearly, the summary was revised as follows (lines 338~352).

“In summary, the present study provided an informative and quantitative big data set relating bacterial growth (population dynamics) to environmental factors, as a successful example of a combination of high-throughput data generation and machine learning. Using a simple ML model to evaluate three growth parameters was likely sufficient to capture the bacterial population dynamics in well-controlled conditions. The differentiation in decision-making components for the lag, exponential and stationary phases protected the bacterial population against extinction. This finding revealed a common and simple strategy of risk diversification for bacterial growth in conditions of excessive resources or starvation, which is a reasonable approach in evolution and ecology. As a representative demonstration, this study showed that investigating the microbial world by data-driven approaches allows us to perceive highly intriguing insights that were inconceivable by traditional biological experiments. Nevertheless, the ML-assisted approach remains as an emerging technology and is required to improve its biological reliability and accessibility for common applications in the studies of life science, microbiology and ecology.”

2) PCA plot and text show that there is an association between media components and E. coli growth. Isn't that obvious?

Yes. According to the statistical significance (p values) of correlation coefficients, the PCA plots indicated the medium combinations were associated with the growth. This observation was likely to promise us that the further analyses would lead to some novel findings of growth law. In accordance, the following revision was made (lines 97~99).

“The results presented an overview of the relationship between the medium combinations and bacterial growth and indicated the growth law in common mediated by the medium components.”

3) 6 different off-shelf ML models are used for the regression and one was chosen for interpretation. That's fine but I do not see the innovation here. It say like to say doing PCR to amplify a gene or CRISPR for a deletion. Off-the-shelf ML methods are now established protocols to run classification/regression when the predictor matrix is very big. What is new here? The application to E. coli growth curve inferred parameters?

The motivation to compare the ML models is to provide the reason (evidence) of why the ML model was chosen. As the ML approaches are not popular (minor methodologies) in biological studies, an appropriate ML model is crucial for the proper and reasonable interpretation of its outcome (result). Whether the target genomic fragment (e.g., gene) is properly amplified by PCR or deleted by CRISPR can be easily and clearly visualized with biological experiments, e.g., electrophoresis or sequencing, because we know the correct answer, e.g., the length or the sequence of the gene. On the other hand, the ML can always give us the output, neither positive nor negative, because we don’t know the correct answer. A preliminary evaluation of different ML models could help us to choose the right one for the analysis and prediction with the experimental dataset. In the present study, the ML model was used to find the components of highest contributions to bacterial growth. The novelty is not the ML model itself but the output of the ML. We are sorry for the insufficient interpretation, and sincerely hope you could allow us to remain the contents in the manuscript. To clarify the purpose of ML selection, the revision was made as follows (lines 103~112).

“As a preliminary test, five representative ML models and an ensemble model were trained and evaluated. The results showed that the prediction accuracy was approximately equivalent among the six ML models (Figure 3B), independent of the evaluation metrics (Figure 3−figure supplement 1). It indicated that the simple ML models were available to tackle the large dataset generated by the throughput growth assay and appropriate for the prediction of bacterial growth according to the environmental details, e.g., medium composition. To determine an explainable linkage between bacterial growth and the medium constitution, the ML model of the gradient-boosted decision tree (GBDT) was chosen for the further investigation.”

4) Growth decision-making. Is this term established and defined in the field. Here it is not explained. Also, who is deciding, the bacteria? Bacteria do not decide.

Sorry for the confusing term usage. The related terms were changed to "decision-making components for bacterial growth" in the revision (lines 150, 204).

5) Sensitivity analysis section is not clear and I especially do not understand how the authors arrive at the final claim (sentence) in that section.

Sorry for the unclear description. To access the analysis of sensitivity in an intuitive way, a schematic drawing illustrating the analytical procedure was newly made and was supplied as the supplementary figure in the revision. The final claim on the participation of Ile (and/or BCAA) in all growth phases was drawn from the finding of the high sensitivity of Ile to the three growth parameters (Figure 5D). The high sensitivity is represented by the large *S*, which is a quantitative value correlated to the changes in growth parameters (*τ*, *r*, and *K*) in response to the concentration gradient of Ile in different combinations of 40 remaining chemicals (i.e., the patterns in Figure 5−figure supplement 1). In addition, the smallest variation was observed among the three *S* in respective to *τ*, *r*, and *K* in Ile (Figure 5E), indicating the highly equivalent responsivity among the three growth parameters. Taken together, Ile was proposed as a global coordinator for bacterial growth. Taking account of the comment, the revision was made as follows (lines 154-165).

“In particular, Ile was the most sensitive component, as it presented the largest *S_g_*, that is, the largest changes in bacterial growth responding to the concentration gradient of Ile in different combinations of other chemicals (i.e., patterns shown in Figure 5−figure supplement 1). The result well supported the finding that the components dominating the three second-priority parameters were Leu and Ile (Figure 4A). Additionally, analysing the variance (*V_s_*) of the *S_τ_, S_r_* and *S_K_* values showed that the largest *V_s_* values was in Cys, (Figure 5F, Figure 5−figure supplement 5), indicating biased sensitivity for the three growth phases. In addition, the smallest *V_s_* was detected in Ile, suggesting equivalent sensitivity all growth phases. Taken together, Ile and/or branched-chain amino acids (BCAAs) participate commonly in all growth phases and are probably global coordinators for bacterial growth.”

6) Line 165-167. I do not understand the sentence. Are the authors claiming S (sulphur) to be the most abundant material on earth? What is the relationship with the results of the FBA modeling? I do not find a connection or at least it is not well explained.

Sorry for the unclear description. We tried to discuss the biological function or mechanism of sulfur in living organisms on earth. The FBA demonstrated the negative correlation between the growth rate and the uptake rate of SO_4_^2-^ (Figure 6B), which was consistent with the finding of that SO_4_^2-^ was the decision-making component for growth rate (Figure 4B, Figure 6A). To achieve biological understanding, the abundance of S on earth and its reactivity in living organisms are discussed. The corresponding sentences were revised as follows (lines 178~183).

“The determinative role of S (SO_4_^2-^) in the growth rate might be related to its function as a material, because it is not only a major constituent of the earth but also the major element in organisms^51, 52, 53^. Since S (SO_4_^2-^) was a highly reactive chemical, e.g., exposure to SO_4_^2-^ increased Reactive Oxygen Species (ROS) levels in bacteria^54^, its determinative role in growth rates was probably mediated by the stress response.”

7) The paper lacks validation. External validation. The authors make claims about bacteria being able to avoid extinction etc. Prove it. The authors are using E. coli growth curves one of the easiest and quickest experimental systems available. Would be great if they could experimentally validate what they claim. In this shape, it feels a bit of an elaborate way to analyse growth curve data.

To valid the ML-mediated prediction, the simulation of population dynamics and the FBA were performed. Both outputs well agreed with the ML prediction, indicating the conclusion was plausible in the viewpoints of growth and metabolism. We assume that the experimental demonstration of the simulated results is requested here. We are sorry to say that it’s not available to design an experiment properly for the reliable demonstration at the moment. To compensate this limitation, the simulation considering the correlations of the growth parameters was additionally performed, and the result was newly provided as Figure 7B. As this simulation takes the relationships of the growth phases into account, it is supposed to be more biological plausible than that shown in Figure 5C. The additional description was made as follows (lines 206~217).

“However, the three parameters remained significantly correlated (Figure 7A), which indicated that the risk diversification strategy did not disturb the trade-off or coordination, e.g., *K/r* selection^59^. The correlations demonstrated that *τ*, *r* and *K* were highly dependent, which well explained why the multimodal distributions of the growth parameters led to only four PCA clusters (Figure 2). Considering the dependency among the three growth parameters, which were decided by three different chemicals (Figure 6A), every 10,000 simulations of population dynamics considering the correlation coefficients (Figure 7A) was additionally performed. The results showed that the three decision-makers facilitated the larger population size than the single decision-maker did (Figure 7B), revealing that the differentiation in decision-making chemicals benefited the bacteria in maintain the final population size.”

In addition, the biological availability was additionally discussed by as follows (lines 196~201).

“In addition, the simulated result of reduced risk of extinction mediated by the differentiation in decision-makers seemed to be biologically reasonable. The previous studies observed the trophic^56^ and functional^57^ redundancy in ecosystems and the genetic redundancy in living cells^58^, which demonstrated that the survivability was secured by the participation of multiple factors. It must be beneficial for maintaining the robustness of ecosystems and cells.”